# Risk factors for human infection with mpox among the Mexican population with social security

Alfonso Vallejos Parás[1], Lumumba Arriaga Nieto[1]*, David Alejandro Cabrera Gaytán[2], Bernardo Cacho Díaz[3], Leticia Jaimes Betancourt[4], Porfirio Felipe Hernández Bautista[2], Oscar Cruz Orozco[1‡], Gabriel Valle Alvarado[1‡], Alejandro Moctezuma Paz[5‡], Mónica Grisel Rivera Mahey[1‡]

1 Epidemiologic Surveillance Coordination, Mexican Institute of Social Security, Mexico City, Mexico, 2 Quality of Supplies and Specialized Laboratories Coordination, Mexican Institute of Social Security, Mexico City, Mexico, 3 Neuroscience Unit, National Cancer Institute, Mexico City, Mexico, 4 Family Medical Unit No 7, Mexican Institute of Social Security, Mexico City, Mexico, 5 Health Research Coordination, Mexican Institute of Social Security, Mexico City, Mexico

☯ These authors contributed equally to this work.
‡ OCO, GVA, AMP and MGRM also contributed equally to this work.
* lumumba.arriaga@imss.gob.mx

**Data Availability Statement:** The data underlying the results presented in the study are available at

## Abstract

### Background

The 2022 mpox outbreak marked a significant shift in the epidemiology of this zoonotic disease, traditionally confined to Central and West Africa. With over 80 countries reporting cases, this outbreak was characterized by a rapid spread in non-endemic regions, leading to more than 70,000 confirmed cases globally.

### Objective

To quantify the cumulative incidence of mpox and identify associated factors of mpox among the Mexican population affiliated by the Mexican Social Security Institute during the 2022–2023 outbreak.

### Material and methods

A retrospective observational study using a cross-sectional survey to assess the cumulative incidence and factors associated with mpox. The cumulative incidence of laboratory confirmed mpox cases was calculated by dividing the number of confirmed cases by the insured population in the Mexican Social Security Institute as of mid-2022, per 100,000 individuals, this was analyzed by sex, age group, sexual orientation and people living with HIV. Logistic regression analyses were conducted to identify sociodemographic and clinical factors associated with mpox infection.

the following Figshare link: https://figshare.com/articles/dataset/Database_mpox_xls/25481266.

**Funding:** The author(s) received no specific funding for this work.

**Competing interests:** The authors have declared that no competing interests exist.

## Results

A total of 2,956 probable cases were reported, with 1,744 (59%) laboratory-confirmed mpox cases. Most confirmed cases were male, with a median age of 32 years. The overall cumulative incidence was 4.05 per 100,000 persons, significantly higher in men and men who have sex with men. Logistic regression revealed that male sex was significantly associated with higher odds of laboratory-confirmed mpox. Age groups 30–34, 35–39, and 40–44 had an elevated risk of infection. Men who have sex with men showed a substantially increased likelihood of mpox, while individuals living with HIV were at higher risk compared to those without HIV. Key clinical predictors included fever, arm rash, and inguinal lymphadenopathy.

## Conclusion

The 2022 mpox outbreak revealed significant disparities in infection risk, particularly among men, men who have sex with men, and individuals living with HIV.

## Introduction

Mpox is an emerging zoonotic infectious disease caused by the mpox virus, a human pathogen belonging to the Orthopoxvirus genus, which also includes the vaccinia and variola viruses, the latter being the causative agent of smallpox [1, 2]. Phylogenetic analyses of the mpox virus reveal two main clades: Clade I, found in the Congo Basin, which has a higher mortality rate of up to 10% and is primarily transmitted by rodents with limited human-to-human spread, and Clade II, which is further subdivided into Clade IIa, occurring in West Africa with low mortality and zoonotic transmission, and Clade IIb, currently spreading globally through human transmission [3, 4]. Recently, a highly divergent Clade I virus, designated Clade Ib, was identified in South Kivu province, further highlighting the ongoing evolution of the virus [5].

The mpox virus was first identified in 1958 in Denmark during investigations of two outbreaks of a smallpox-like disease in cynomolgus monkeys [6]. The first human case of mpox was reported in 1970 in the Democratic Republic of Congo (DRC) [7]. For decades, mpox cases remained largely confined to forested areas of Central and West Africa, with sporadic outbreaks primarily occurring in the DRC [8, 9]. In countries across Central and West Africa, outbreaks of monkeypox have consistently occurred in populations living in rural areas, in small villages (less than 1000 people) adjacent to or within humid evergreen tropical forests–at the so-called human-animal interface [8].

In 2003, the first cases of mpox outside Africa were reported during an outbreak in Wisconsin, USA, involving 11 individuals who developed febrile illnesses with skin eruptions after direct contact with infected prairie dogs. Epidemiological investigations suggested that the prairie dogs had been exposed to rodents recently imported from West Africa [10]. Subsequent travel-associated cases were reported in Israel in 2018, the UK in 2018 and 2019, and Singapore in 2019, all linked to exposures in Nigeria [11].

A significant shift occurred in May 2022, when mpox cases were reported in countries where the disease was not previously endemic, primarily affecting men who have sex with men (MSM) and detected through sexual health services. On July 23, 2022, the World Health Organization (WHO) declared the mpox outbreak a Public Health Emergency of International Concern (PHEIC) due to its rapid spread, with over 16,000 cases reported across 75 countries

and territories at that time in the 2022 multinational mpox outbreak [12, 13], the disease was described clinically with fever, lethargy, myalgia, headache, lymphadenopathy, and skin rash. The skin lesions, often located in the anogenital area, trunk, arms, legs, face, palms, and soles, can be macular, pustular, vesicular, or crusted, appearing in multiple phases simultaneously. The number of lesions varies widely, typically with ten or fewer per individual [14, 15]. The mean incubation period ranges from 5.6 days for symptom onset to 7.5 days for rash onset [16]. Hospitalization rates are around 14.1%, with significant variability across countries and time, and the estimated case fatality rate is 0.03% [17].

In Mexico, the first mpox case was registered in May 2022, and according to the 2023 last official report of the Ministry of Health, up to May 31, 2023, there had been 6,864 persons identified as probable mpox cases. Of these, 4,021 were confirmed by laboratory testing, 238 were under investigation, and 2,605 had been excluded by laboratory testing [18].

This global health emergency was declared over in May 2023 after a sustained decline in global cases [19].

On 14 August 2024, the WHO Director-General, based on the recommendation of the International Health Regulations Emergency Committee, determined that the increase in mpox cases in the Democratic Republic of the Congo and a growing number of countries in Africa constitutes a new PHEIC, with the potential to spread further across African countries and possibly beyond the continent [20].

This study aimed to quantify the cumulative incidence of mpox and identify associated factors of mpox among the Mexican population affiliated by the Mexican Social Security Institute during the 2022–2023 outbreak.

## Materials and methods

We conducted a retrospective observational study using a cross-sectional survey to assess mpox cumulative incidence and associated factors in a population with health coverage from the Mexican Social Security Institute (IMSS).

The Mexican health system consists of three main components operating in parallel: 1) employment based social insurance schemes, 2) public assistance services for the uninsured, and 3) a private sector [21]. The IMSS is the largest social security institution in Latin America [22], providing social insurance to private sector employees [21]. The institution provides full healthcare coverage to its affiliates: there are no restrictions on service utilization, requirements for copayment or predefined limits on coverage [23]. Recent. In 2022, the IMSS registered more than 60 million beneficiaries [24], which corresponds to 46.3% of the Mexican population [25].

The data were obtained from the Mpox Epidemiological Surveillance System, which operates nationwide across all medical units in the country. Consequently, medical units within the IMSS are mandated to report all cases that meet the criteria for probable mpox cases to the National Epidemiological Surveillance System for Mpox [26]. Through laboratory surveillance, each case is investigated and classified according to the operational definitions outlined below. All cases reported to the mpox epidemiological surveillance system and studied in this research are new cases of the disease.

### Variable definitions and data collection

The operational definition of a probable case of mpox to be reported in the epidemiological surveillance system was any person of any age and sex who has one or more lesions in the skin (macule, papule, vesicle, pustule, and/or scab) or mucosal surface without other clinical diagnosis explaining the clinical presentation, and who has one or more of the following signs or

symptoms: fever, myalgias, headache, lymphadenopathy, asthenia, arthralgia, low back pain. In immunocompromised individuals, a probable case was defined as having one or more skin or mucosal surface lesions, even if no other signs or symptoms were present.

Also, patients without identifiable skin or mucosal lesions but who presented one or more of the following signs or symptoms: headache, acute onset fever ($> 38.5°C$), lymphadenopathy, myalgia, low back pain or asthenia, and were in contact with a case confirmed during the 21 days before the onset of symptoms, were considered as a probable case of mpox.

Laboratory-Confirmed Case of Mpox: A probable case with a positive sample result for the mpox virus processed by approved laboratories for diagnosis.

Laboratory Discarded Case of Mpox: A probable case with a negative result for the mpox virus processed by approved laboratories for diagnosis, without evidence of clinical or epidemiological association.

All probable cases identified in IMSS medical units were notified and sociodemographic, epidemiological, and clinical variables were collected using a case report form (CRF) standardized by the mpox National Epidemiological Surveillance System [26]. The CRF was collected through the physician's interview with the patient so the responses to the questionnaire are collected directly from the patient and medical evaluation.

The variables analyzed in the CRF included sex assigned at birth (male or female) and sexual orientation, which was self-reported. The categories of sexual orientation considered were heterosexual, MSM, bisexual, and lesbian. Gender was also self-reported, with the following categories: cisgender women, cisgender men, transgender women, and non-binary individuals. Health conditions such as HIV, syphilis, hepatitis C, and other sexually transmitted infections (STIs) were recorded through self-reporting.

Additionally, data were collected on the patient's clinical status, including whether they were hospitalized or deceased, based on medical reports. Sociodemographic and clinical variables were also recorded, such as the presence of fever, exanthema, conjunctivitis, odynophagia, ulcers, and lymphadenopathy.

All patients classified as probable cases were sampled for PCR confirmation. Samples were processed for molecular detection of the mpox DNA by polymerase chain reaction (PCR) tests performed by laboratories authorized by the Institute of Epidemiological Diagnosis and Reference (InDRE), the national reference laboratory in the country. The laboratory specimens included exudate smears from skin lesions (vesicles or pustules) or mucosal surface exudates (oropharyngeal or pharyngeal).

## Study population

All probable mpox cases reported by IMSS medical units from May 2022 to May 2023 involving individuals aged 15 years and older were included in the study. Cases were excluded if laboratory results were unavailable during the study period, including those rejected due to specimen quality, indeterminate results, or samples that had not yet been processed (Fig 1). Database was accessed on June 26, 2023.

## Statistical analysis

A descriptive analysis of the sociodemographic and clinical variables of patients confirmed with mpox by laboratory testing was conducted. The variables included age, sex at birth, gender, sexual orientation, pre-existing health conditions, comorbidities, hospitalization, and mortality.

A comparative analysis was performed to evaluate the clinical symptoms and signs between patients with laboratory-confirmed mpox and those with laboratory-discarded mpox. The

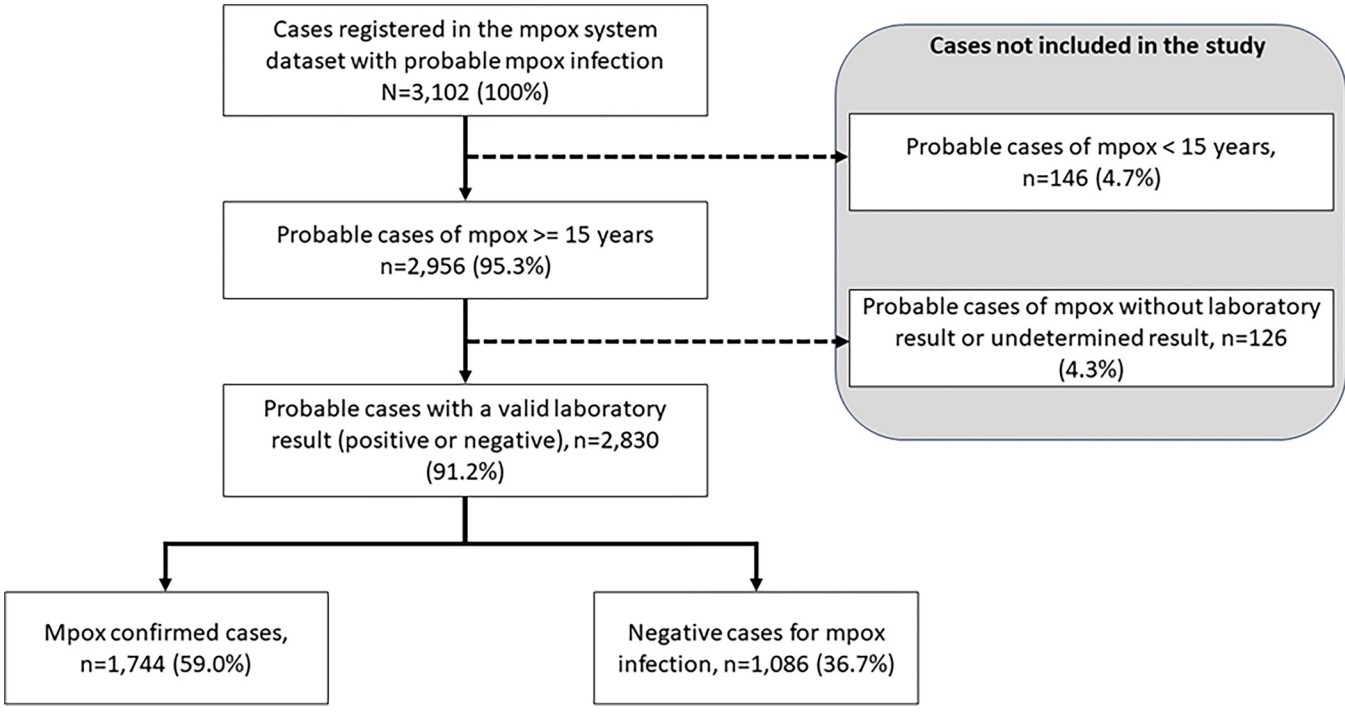

**Fig 1. Flow chart of sample composition of the study.**

analysis included a wide range of symptoms, such as fever, rash, lymphadenopathy, and pain at various anatomical sites. For each variable, the presence or absence of symptoms was recorded, and the percentage of patients in each group was calculated. The chi-squared test was used to assess the statistical significance of differences between the two groups, with a significance level of $p < 0.05$.

An epidemic curve of mpox cases by month was constructed using the number of laboratory-confirmed cases and discarded cases from May 2022 to May 2023. The monthly positivity rate was calculated by dividing the number of confirmed mpox cases by the total number of cases tested (confirmed and discarded) and expressing the result as a percentage.

The cumulative incidence was calculated by dividing the number of laboratory-confirmed cases of mpox among the population insured in the IMSS at mid-2022 per 100,000 individuals [27]. The cumulative incidence was obtained by sex and age group.

The cumulative incidence of laboratory-confirmed cases of mpox was also obtained in population subgroups, by sexual orientation, MSM and people living with HIV per 100,000 individuals.

Based on data from the 2021 National Survey on Sexual and Gender Diversity conducted by the National Institute of Statistics and Geography (INEGI) [28], the incidence of population subgroups by sexual orientation was calculated. The proportions of individuals aged 15 and older identifying as LGB+ in Mexico were used, with 4.2% for males and 5.3% for females. Additionally, 26.5% of MSM were considered within the total population of men identifying as LGB+. These estimates were applied to the population affiliated with the IMSS and used as the denominator to calculate the cumulative incidence of mpox in these population subgroups.

For the calculation of the cumulative incidence of mpox in the population subgroup of people living with HIV, official IMSS reports were used as the denominator, with a total of 82,716 people living with HIV receiving medical care [29].

A bivariate analysis was conducted to assess the association between each variable and Mpox infection. The analysis generated odds ratios (OR) along with 95% confidence intervals (CI), and statistical significance was determined using p-values. The Chi-square test was employed as the hypothesis testing method, with a significance level set at p < 0.05. S1 Table.

A multivariate analysis was conducted using binary regression to evaluate the association between sociodemographic and clinical variables and the diagnosis of a laboratory-confirmed case of mpox. The binary regression model incorporated independent variables that exhibited statistical significance in the preceding univariate analysis.

The Wald statistic was used to assess the statistical significance of individual predictors within the model. The odds ratio (OR) was utilized as a measure of association, along with 95% confidence intervals. The model's explanatory power was measured using Nagelkerke R-squared. To evaluate the model's goodness-of-fit, the Hosmer and Lemeshow test was performed. A significance level of p < 0.05 was established.

Statistical analyses were performed using SPSS software and Excel. The data are open and can be accessed at the following URL: https://figshare.com/articles/dataset/Database_mpox_xls/25481266

## Ethical considerations

The Institute's Research and Ethics Committee approved the study with the registration number R-2023-3605-016. Informed consent was not needed because all data were fully anonymized before access. All authors did not have access to information that could identify individual participants during or after data collection. The recommendations for ethical surveillance of mpox issued by the Health Organization were considered [30].

## Results

A total of 2,956 probable cases ≥15 years old were reported in the surveillance system, and 59% (n = 1,744/2,956) were laboratory-confirmed cases of mpox.

### Characteristics of the laboratory-confirmed mpox cases (Table 1)

Among 1,744 laboratory-confirmed Mpox cases, the median age was 32 years (IQR: 22–42), with individuals ranging from 15 to 84 years. The most common age groups were 30–34 years (26.1%) and 25–29 years (23.3%). Regarding sex assigned at birth, the majority were male (96.0%), with 4.0% being female. In terms of gender identity, 88.6% were cisgender men and 4.2% were cisgender women, while 0.8% identified as non-binary, and 6.4% were unknown or unregistered. Concerning sexual orientation, 70.0% identified as MSM, while 17.9% were heterosexual and 7.2% bisexual. 56.0% of individuals were living with HIV, while 44.0% were not. Additionally, 3.3% had syphilis, and 1.5% had hepatitis C. Most patients, 94.2%, were not hospitalized, though 5.8% required hospitalization. Finally, the mortality rate was 0.7%, with 99.3% of cases surviving.

The clinical findings are presented in Table 2. The most frequent symptoms were fever (80.6%), rash (99.7%), and lymphadenopathies (59.9%). Rashes were present in nearly all cases, both confirmed and discarded. Lesions were most observed on the arms (74.7%), trunk (69.4%), and anogenital region (58.1%) among confirmed cases. Inguinal lymphadenopathies were also notably more common in confirmed cases (32.3%) compared to discarded ones (9.4%).

Several symptoms displayed statistically significant differences between confirmed and discarded cases: fever was more prevalent in confirmed cases (80.6% vs. 69.3%), anogenital lesions were more frequent in confirmed cases (58.1% vs. 25.7%), and lymphadenopathies were

**Table 1. Demographic and clinical characteristics of laboratory-confirmed mpox cases.**

| Variable | Categories | Laboratory-confirmed mpox case | |
|---|---|---|---|
| | | n = 1,744 | |
| | | Count | % |
| Age | median: (IQR), full range | 32 (22–42) 15–84 | 100% |
| Age group | 15–19 | 24 | 1.4% |
| | 20–24 | 177 | 10.1% |
| | 25–29 | 407 | 23.3% |
| | 30–34 | 456 | 26.1% |
| | 35–39 | 300 | 17.2% |
| | 40–44 | 182 | 10.4% |
| | 45–49 | 110 | 6.3% |
| | 50–54 | 48 | 2.8% |
| | 55–59 | 28 | 1.6% |
| | 60–64 | 6 | 0.3% |
| | ≥65 | 6 | 0.3% |
| Sex assigned at birth | Female | 69 | 4.0% |
| | Male | 1,675 | 96.0% |
| Gender | Cisgender women | 73 | 4.2% |
| | Cisgender men | 1,546 | 88.6% |
| | Non-binary | 14 | 0.8% |
| | Transgender women | 0 | 0.0% |
| | Unknown/ Not registered | 111 | 6.4% |
| Sexual orientation | Heterosexual | 313 | 17.9% |
| | MSM | 1,220 | 70.0% |
| | Bisexual | 125 | 7.2% |
| | Lesbian | 2 | 0.1% |
| | Other | 5 | 0.3% |
| | Unknown/ Not registered | 79 | 4.5% |
| People living with HIV | No | 768 | 44.0% |
| | Yes | 976 | 56.0% |
| Syphilis | No | 1,687 | 96.7% |
| | Yes | 57 | 3.3% |
| Hepatitis C | No | 1,718 | 98.5% |
| | Yes | 26 | 1.5% |
| Hospitalized | No | 1,643 | 94.2% |
| | Yes | 101 | 5.8% |
| Died | No | 1,732 | 99.3% |
| | Yes | 12 | 0.7% |

IQR = interquartile range

observed more often in confirmed cases (59.9% vs. 38.4%). Painful ulcers were also more common in confirmed cases (16.8%) compared to discarded cases (12.9%).

Symptoms like myalgias, headaches, and articular pain appeared at similar rates across both confirmed and discarded cases.

Fig 2 shows epidemiological curve of laboratory-confirmed Mpox cases with positivity rates according to the month of symptom onset. The first mpox case occurred in May 2022, and the

**Table 2. Comparison of clinical symptoms between laboratory-confirmed and laboratory-discarded mpox cases.**

| Variable | Categories | Laboratory confirmed case of mpox | | Laboratory discarded case of mpox | | p value |
|---|---|---|---|---|---|---|
| | | n = 1,744 | | n = 1,086 | | |
| | | Count | % of column | Count | % of column | |
| Fever | Yes | 1406 | 80.6% | 753 | 69.3% | 0.000 |
| | No | 338 | 19.4% | 333 | 30.7% | |
| Rash (skin/mucosal lesions) | Yes | 1739 | 99.7% | 1082 | 99.6% | 0.708 |
| | No | 5 | 0.3% | 4 | 0.4% | |
| • Head | Yes | 671 | 38.5% | 424 | 39.0% | 0.763 |
| | No | 1073 | 61.5% | 662 | 61.0% | |
| • Face | Yes | 1050 | 60.2% | 602 | 55.4% | 0.012 |
| | No | 694 | 39.8% | 484 | 44.6% | |
| • Neck | Yes | 562 | 32.2% | 419 | 38.6% | 0.001 |
| | No | 1182 | 67.8% | 667 | 61.4% | |
| • Oral cavity | Yes | 290 | 16.6% | 181 | 16.7% | 0.979 |
| | No | 1454 | 83.4% | 905 | 83.3% | |
| • Arms | Yes | 1302 | 74.7% | 719 | 66.2% | 0.000 |
| | No | 442 | 25.3% | 367 | 33.8% | |
| • Legs | Yes | 979 | 56.3% | 612 | 56.6% | 0.909 |
| | No | 765 | 43.7% | 474 | 43.4% | |
| • Trunk | Yes | 1210 | 69.4% | 731 | 67.3% | 0.249 |
| | No | 534 | 30.6% | 355 | 32.7% | |
| • Palms/soles | Yes | 643 | 36.9% | 402 | 37.0% | 0.937 |
| | No | 1101 | 63.1% | 684 | 63.0% | |
| • Anogenital region | Yes | 1013 | 58.1% | 279 | 25.7% | 0.000 |
| | No | 731 | 41.9% | 807 | 74.3% | |
| Lymphadenopathies | Yes | 1044 | 59.9% | 417 | 38.4% | 0.000 |
| | No | 700 | 40.1% | 669 | 61.6% | |
| • L. Cervical | Yes | 592 | 33.9% | 328 | 30.2% | 0.039 |
| | No | 1152 | 66.1% | 758 | 69.8% | |
| • L. Axillary | Yes | 122 | 7.0% | 86 | 7.9% | 0.360 |
| | No | 1622 | 93.0% | 1000 | 92.1% | |
| • L. Inguinal | Yes | 563 | 32.3% | 102 | 9.4% | 0.000 |
| | No | 1181 | 67.7% | 984 | 90.6% | |
| Headache | Yes | 1222 | 70.1% | 729 | 67.1% | 0.100 |
| | No | 522 | 29.9% | 357 | 32.9% | |
| Articular pain | Yes | 940 | 53.9% | 587 | 54.1% | 0.937 |
| | No | 804 | 46.1% | 499 | 46.0% | |
| Myalgias | Yes | 1157 | 66.3% | 699 | 64.4% | 0.282 |
| | No | 587 | 33.7% | 387 | 35.6% | |
| Cough | Yes | 290 | 16.6% | 181 | 16.7% | 0.979 |
| | No | 1454 | 83.4% | 905 | 83.3% | |
| Nausea | Yes | 208 | 11.9% | 147 | 13.5% | 0.209 |
| | No | 1536 | 88.1% | 939 | 86.5% | |
| Vomit | Yes | 72 | 4.1% | 48 | 4.4% | 0.708 |
| | No | 1672 | 95.9% | 1038 | 95.6% | |
| Odynophagia | Yes | 800 | 45.9% | 445 | 41.0% | 0.011 |
| | No | 944 | 54.1% | 641 | 59.0% | |

(*Continued*)

**Table 2.** (Continued)

| Variable | Categories | Laboratory confirmed case of mpox | | Laboratory discarded case of mpox | | p value |
|---|---|---|---|---|---|---|
| | | n = 1,744 | | n = 1,086 | | |
| | | Count | % of column | Count | % of column | |
| Conjunctivitis | Yes | 109 | 6.3% | 70 | 6.5% | 0.835 |
| | No | 1635 | 93.7% | 1016 | 93.5% | |
| Lower back pain | Yes | 526 | 30.2% | 294 | 27.1% | 0.078 |
| | No | 1218 | 69.8% | 792 | 72.9% | |
| Bleeding ulcers | Yes | 79 | 4.5% | 37 | 3.4% | 0.143 |
| | No | 1665 | 95.5% | 1049 | 96.6% | |
| Painful ulcers | Yes | 293 | 16.8% | 140 | 12.9% | 0.005 |
| | No | 1451 | 83.2% | 946 | 87.1% | |

*All variables were tested with Pearson's Chi-square Test

number of cases continued to increase until a peak in September 2022, with n = 488 laboratory-confirmed Mpox cases; since then, a sustained decrease in the number of cases has been observed. The highest positivity rate was in September 2022 at 73.6%. The positivity rate showed a general decline from April 2023 onwards, with the lowest in April (9.8%) and a slight increase in May (13.3%).

## Cumulative incidence

Table 3 shows that the incidence rates of laboratory-confirmed Mpox cases vary significantly across age groups, with higher rates observed in younger cohorts. Notably, the 30 to 34 age group exhibits the highest incidence rate, reaching 21.98 per 100,000 IMSS beneficiaries

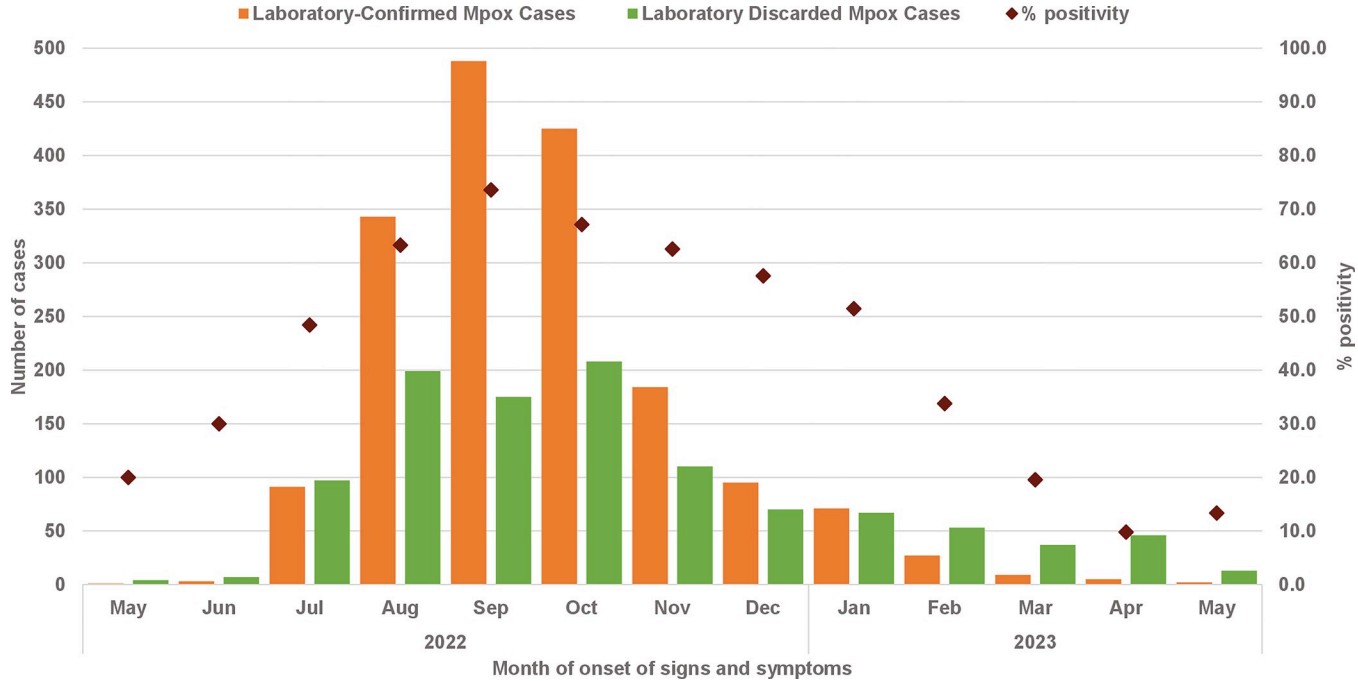

**Fig 2. Epidemiological curve of laboratory-confirmed mpox cases and positivity rates (may 2022—may 2023) among IMSS beneficiaries.**

**Table 3. Cumulative incidence of laboratory-confirmed mpox cases by sex assigned at birth and age group (2022–2023) per 100,000 IMSS beneficiaries.**

| Age group | Population (Male) | Population (Female) | Laboratory-Confirmed Mpox Cases (Male) | Laboratory-Confirmed Mpox Cases (Female) | Cumulative Incidence (Male) per 100,000 population | Cumulative Incidence (Female) per 100,000 population |
|---|---|---|---|---|---|---|
| 15 to 19 | 1,153,925 | 1,222,650 | 20 | 4 | 1.73 | 0.33 |
| 20 to 24 | 1,994,684 | 2,005,068 | 166 | 11 | 8.32 | 0.55 |
| 25 to 29 | 2,204,177 | 2,251,938 | 394 | 13 | 17.88 | 0.58 |
| 30 to 34 | 2,011,117 | 2,188,668 | 442 | 14 | 21.98 | 0.64 |
| 35 to 39 | 1,813,102 | 2,104,708 | 295 | 5 | 16.27 | 0.24 |
| 40 to 44 | 1,687,138 | 2,019,692 | 177 | 5 | 10.49 | 0.25 |
| 45 to 49 | 1,639,375 | 2,027,962 | 103 | 7 | 6.28 | 0.35 |
| 50 to 54 | 1,463,244 | 1,871,547 | 41 | 7 | 2.8 | 0.37 |
| 55 to 59 | 1,294,288 | 1,703,431 | 26 | 2 | 2.01 | 0.12 |
| 60 to 64 | 1,232,823 | 1,600,708 | 6 | 0 | 0.49 | 0 |
| 65 and over | 3,297,141 | 4,315,821 | 5 | 1 | 0.15 | 0.02 |
| **Total** | **19,791,014** | **23,312,193** | **1,675** | **69** | **8.46** | **0.3** |

among males and 0.64 among females. In contrast, the 60 to 64 age group exhibits a much lower incidence rate, with 0.49 per 100,000 for males and 0 for females.

Table 4 outlines the cumulative incidence during the study period based on sex at birth, sexual orientation, and HIV status. We found a cumulative incidence of 4.05 cases per 100,000 people overall. The incidence among males was 8.46 cases per 100,000, while for females, it was much lower at 0.30 per 100,000. Heterosexual individuals (both male and female) had a cumulative incidence of 0.76 cases per 100,000, while non-heterosexual individuals (including MSM, bisexual, and lesbian) had a significantly higher incidence of 65.42 cases per 100,000. Within this group, MSM had the highest incidence, with 553.86 cases per 100,000. Among people living with HIV, the cumulative incidence rate was 1,179.94 per 100,000 persons, compared to 1.79 per 100,000 for those not living with HIV.

To determine the relationship between each potential predictor variable and the presence of mpox individually, we performed a bivariate analysis which is available in S1 Table.

Being male was strongly associated with laboratory-confirmed mpox cases: OR 10.87 (95% CI 8.28–14.29), p<0.001; the age groups with the greatest strength of association with the disease were 30 to 34 years OR 12.44 (95% CI 5.06–30.58), p<0.001; 35 to 39 years OR 13.59

**Table 4. Laboratory-confirmed mpox cases and cumulative incidence by sex assigned at birth, sexual orientation, and HIV status per 100,000 IMSS beneficiaries.**

| Categories | Laboratory-confirmed mpox cases | Population | Cumulative incidence (per 100,000) |
|---|---|---|---|
| Total population | 1,744 | 43,103,207 | 4.05 |
| Female | 69 | 23,312,193 | 0.30 |
| Male | 1,675 | 19,791,014 | 8.46 |
| Heterosexual (female and male) | 313 | 41,036,438 | 0.76 |
| Non-heterosexual (including MSM, bisexual and lesbian) | 1,352 | 2,066,769 | 65.42 |
| Heterosexual (male) | 250 | 19,570,740 | 1.28 |
| MSM* | 1,220 | 220,274 | 553.86 |
| People not living with HIV | 768 | 43,020,491 | 1.79 |
| People living with HIV | 976 | 82,716 | 1179.94 |

*Men who have sex with men

(95% CI 5.47–33.75), p<0.001; and 40 to 44 years OR 12.87 (95% CI 5.10–32.47), p<0.001. Fever was associated with laboratory-confirmed cases OR 1.84 (95% CI 1.54–2.19), p<0.001; rash in the anogenital region was strongly associated laboratory-confirmed mpox cases OR 4.01 (95% CI 3.40–4.73), p<0.001; similarly, inguinal lymphadenopathy OR 4.60 (95% CI 3.66–5.77), p<0.001. Other sociodemographic and clinical characteristics, as well as the results of the bivariate analysis are shown in S1 Table.

### Multivariate analysis (Table 5)

Regression was performed with a full sample of 2,830 patients, of which 1,744 (61.6%) had laboratory-confirmed cases of mpox and 1,086 (38.4%) were laboratory-discarded cases of mpox. The multivariate model had a statistical significance of p <0.001 using the Wald statistic, with a Nagelkerke R-squared of 0.485. The Hosmer and Lemeshow test was used to evaluate the model, resulting in a p = 0.083; the model was able to correctly classify 84.6% of mpox cases and 73.1% of non-mpox cases, with an overall prognostic classification percentage of 80.2%. After multivariate analysis, several variables remained statistically associated with a higher possibility of having a confirmed mpox infection. First, sex at birth indicated that males had significantly higher odds OR 3.287 (95% CI 2.445–4.418), p <0.001; age group analysis showed that several age groups were significantly associated with the condition, with older age groups generally having higher odds ratios; these odds ratios increased from ages 15–19 to ages 45–49, ranging from 1.050 to 4.006, with p-values < 0.05 for the significant age groups. In terms of sexual orientation, MSM OR 4.325 (95% CI 3.437–5.442), p < 0.001; health condition also played a role, individuals living with HIV had significantly higher odds OR 2.499 (95% CI

**Table 5. Multivariate analysis of factors associated with laboratory-confirmed cases of mpox: Logistic regression results.**

| Variable | Categories | Sig. | Adjusted Odds Ratio | 95% confidence interval for Odds ratio | |
|---|---|---|---|---|---|
| | | | | Lower | Upper |
| Sex assigned at birth | Female | | 1 | | |
| | Male | < .001 | 2.792 | 2.055 | 3.791 |
| Age group | 15–19 | .932 | 1.050 | .346 | 3.185 |
| | 20–24 | .644 | 1.266 | .466 | 3.436 |
| | 25–29 | .063 | 2.552 | .949 | 6.859 |
| | 30–34 | .015 | 3.406 | 1.263 | 9.181 |
| | 35–39 | .009 | 3.800 | 1.390 | 10.387 |
| | 40–44 | .008 | 4.006 | 1.435 | 11.184 |
| | 45–49 | .021 | 3.432 | 1.209 | 9.739 |
| | 50–54 | .116 | 2.387 | .806 | 7.067 |
| | 55–59 | .187 | 2.184 | .684 | 6.970 |
| | 60–64 | .943 | .947 | .214 | 4.185 |
| | 65 + | | 1 | | |
| Sexual orientation | MSM* | < .001 | 4.325 | 3.437 | 5.442 |
| Health condition | Living with HIV | < .001 | 2.499 | 1.951 | 3.202 |
| Signs and symptoms | Fever | < .001 | 1.560 | 1.242 | 1.959 |
| | Rash in arms | < .001 | 1.524 | 1.229 | 1.890 |
| | Rash anogenital | < .001 | 2.226 | 1.809 | 2.738 |
| | Lymphadenopathy inguinal | < .001 | 2.319 | 1.761 | 3.053 |
| Constant | | < .001 | .034 | | |

*Men who have sex with men

1.951–3.202), p <0.001 compared to those without HIV; finally, signs and symptoms statistically associated with laboratory-confirmed mpox case included fever OR 1.56 (95% CI 1.24–1.95), p <0.001; rash located in arms OR 1.52 (95% CI 1.22–1.89), p <0.001; rash in anogenital region OR 2.22 (95% CI 1.80–2.73), p <0.001; and inguinal lymphadenopathy OR 2.31 (95% CI 1.76–3.05), p <0.001, all compared to patients without these signs and symptoms.

## Discussion

In this study, we investigated the cumulative incidence of laboratory-confirmed mpox cases and its associated factors among the Mexican population covered by the Mexican Social Security Institute during the 2022–2023 outbreak. Our findings show significant differences in incidence rates by sex assigned at birth and age groups. The data also indicated that non-heterosexual individuals, particularly MSM, and people living with HIV, had a high incidence of mpox infection.

Multivariate analysis indicated that being male, identifying as MSM, and people living with HIV were significantly associated with increased odds of laboratory-confirmed mpox cases. Additionally, symptoms such as fever, rash on the arms, and inguinal lymphadenopathy were significant predictors of mpox infection.

Our results on the age of mpox cases are consistent with other studies in the Americas, which report a median age of 32 years [31, 32]. However, this is lower than the median age reported in European countries (39 years) [33, 34], likely due to demographic trends between the Americas and Europe [35]. We found that mpox occurred more frequently in males (96%) than in females (4%), consistent with findings from other countries [32, 36]. The cumulative incidence of mpox in our study was 4.05 per 100,000 persons, direct comparisons are challenging due to the limited availability of incidence rate data for mpox across different countries.

The observed proportion of mpox cases in non-binary persons was 0.8%, lower than the 1.7% reported in the US [37]. Our findings did not establish any significant association between gender and mpox infection. In terms of sexual orientation, we found that the proportion of cases in the MSM group accounted for 70% of confirmed mpox cases. People with a non-heterosexual orientation had an incidence rate of 65.42 per 100,000. The MSM group showed higher associations with mpox infection. The WHO has established that gender-diverse and transgender people are more vulnerable to mpox infection than others [38].

Our study observed a statistically significant association between mpox infection and living with HIV, with a cumulative incidence of 1,179.94 per 100,000 persons. The proportion of mpox cases living with HIV was 56%, higher than the 40.32% reported in a systematic review and other studies [39, 40]. Our results demonstrate a hospitalization rate of 5.2% among confirmed mpox patients, which is close to the 6% reported in other studies [17, 41].

The found mortality rate of mpox was 0.69 per 100 cases, lower than the 0.72 per 100 cases reported by health authorities in Mexico [18], but 5.3 times higher than the rate reported in the United States of America, at 0.13 per 100 confirmed cases [31].

Clinically, confirmed mpox cases presented distinct symptoms, including fever, inguinal lymphadenopathy, and rash in the arms and anogenital region, which were significantly more prevalent compared to non-mpox cases. The monkeypox virus is spread through close skin-to-skin contact; clinically, we found that inguinal lymphadenopathy and rash located in the anogenital region were associated with a higher likelihood of being a confirmed mpox case, consistent with findings from other studies [42, 43]. These findings have been linked to increased mpox transmission through close physical contact during intimate or sexual activity.

This study provides an overview of mpox infection in the medical-insured population of Mexico. Due to the study's observational nature, we must clarify that the information comes

from a self-reported questionnaire answered by individuals participating in the survey, which may introduce bias, and the presence of diseases such as syphilis, or living with HIV has not been confirmed as part of the study. We also excluded suspected cases without laboratory results, accounting for 4.3% of the suspected cases, which may lead to an underestimation of mpox incidence.

We acknowledge that the absence of variables such as socioeconomic status, region or place of residence, and sexual behavior (including the number of sexual partners and casual sexual contacts) constitutes a limitation of our study. These factors could potentially influence the risk of mpox infection.

These findings underscore the importance of targeted public health interventions, and surveillance strategies, particularly for high-risk groups, to mitigate the spread of Mpox infection.

## Supporting information

**S1 Table. Bivariate analysis of variables associated with laboratory-confirmed mpox cases.** (DOCX)

## Acknowledgments

We would like to express our sincere gratitude to the epidemiological surveillance personnel of the Mexican Social Security Institute (IMSS) for their invaluable work and dedication.

## Author Contributions

**Conceptualization:** Alfonso Vallejos Parás, Lumumba Arriaga Nieto, David Alejandro Cabrera Gaytán, Leticia Jaimes Betancourt, Porfirio Felipe Hernández Bautista, Oscar Cruz Orozco, Gabriel Valle Alvarado, Alejandro Moctezuma Paz, Mónica Grisel Rivera Mahey.

**Data curation:** Alfonso Vallejos Parás, Lumumba Arriaga Nieto, Leticia Jaimes Betancourt, Porfirio Felipe Hernández Bautista, Oscar Cruz Orozco, Gabriel Valle Alvarado, Mónica Grisel Rivera Mahey.

**Formal analysis:** Alfonso Vallejos Parás, Lumumba Arriaga Nieto, David Alejandro Cabrera Gaytán, Bernardo Cacho Díaz, Porfirio Felipe Hernández Bautista.

**Investigation:** Alfonso Vallejos Parás, David Alejandro Cabrera Gaytán, Bernardo Cacho Díaz, Leticia Jaimes Betancourt, Oscar Cruz Orozco, Alejandro Moctezuma Paz, Mónica Grisel Rivera Mahey.

**Methodology:** Alfonso Vallejos Parás, Lumumba Arriaga Nieto, David Alejandro Cabrera Gaytán, Bernardo Cacho Díaz, Porfirio Felipe Hernández Bautista, Oscar Cruz Orozco, Gabriel Valle Alvarado, Alejandro Moctezuma Paz.

**Supervision:** Bernardo Cacho Díaz, Leticia Jaimes Betancourt, Oscar Cruz Orozco, Mónica Grisel Rivera Mahey.

**Validation:** Alfonso Vallejos Parás, Lumumba Arriaga Nieto, Bernardo Cacho Díaz, Leticia Jaimes Betancourt, Porfirio Felipe Hernández Bautista, Gabriel Valle Alvarado, Alejandro Moctezuma Paz.

**Writing – original draft:** Alfonso Vallejos Parás, Lumumba Arriaga Nieto.

**Writing – review & editing:** Alfonso Vallejos Parás, Lumumba Arriaga Nieto, Bernardo Cacho Díaz.

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
