## [Decision Letter · Decision Letter 0]

15 Feb 2024

PONE-D-24-01590Risk factors for human infection with Mpox among the Mexican populationPLOS ONE

Dear Dr. Nieto,

Thank you for submitting your manuscript to PLOS ONE. After careful consideration, we feel that it has merit but does not fully meet PLOS ONE’s publication criteria as it currently stands. Therefore, we invite you to submit a revised version of the manuscript that addresses the points raised during the review process.

We look forward to receiving your revised manuscript.

Kind regards,

Abdelaziz Abdelaal, M.D.

Academic Editor

PLOS ONE

Journal Requirements:

Additional Editor Comments:

Thanks to the authors for considering PLOS ONE to publish their research. Based on my evaluation and the reviewers' comments, we believe that the manuscript, in its current form, provides quite similar results to available evidence without any novel findings. Additionally, there were several major limitations that limited the interpretability of the reported findings.

These comments, listed below, require the redoing of the statistical analysis. A consultation with an expert statistician is recommended.

If the authors can address these concerns and modify their manuscript accordingly, I'd be happy to reconsider the manuscript for further evaluation.

The Editor's Comments:

1- Why was the HR caclulated as an epidemiological measure? In your case, the best measure would be the cumulative incidence rate, known as the incidence proportion over the specified time period.

2- Given the large sample size, i would suspect that age be normally distributed. Why was the age presented as median (IQR)? I would advise the authors to provide age in different forms (continuous data, categorical data)

3- There is lack of information on how the regression analysis was attempted? Which variables were included in the univariate model? and how was the selected of the variables imported in the multivariable model made?

4- Was model fit analysis performed? I known that the authors provided the Hosmer and Lemeshow test results; however, there are NOT consistent with the provided R-squared which reflects poor fit (which indicates that the analyzed factors do not reflect the change in the measured outcomes; there are other factors that were not accounted for). This is also supported by the wide confidence interval for variables that have a large sample size, which would normally reflect a narrow CI (like male in the multivariate model).

5- This issue is the most important of all, which is the lack of enough baseline data of included patients. Providing age, gender, sexual orientation, and HIV coinfection is not sufficient. Were not there any other data to be analyzed in your dataset? The interpretation from this dataset, given the scarcity of data, is very limited and not different from that already reported in the literature during the disease outbreak.

6- Was multicollinearity considered? and how was it identified and dealt with?

7- In line 203, diabetes was put in the regression mode, where it was not even mentioned in the baseline variables table (Table 1). Were variables selected on purpose? You need to standardize the reporting of your data.

8- What comorbid conditions were considered in Table 1? numerical data is required. Was HIV considered a part of these data?

Reviewers' comments:

Reviewer's Responses to Questions

**Comments to the Author**

1. Is the manuscript technically sound, and do the data support the conclusions?

Reviewer #1: Partly

Reviewer #2: Yes

2. Has the statistical analysis been performed appropriately and rigorously? 

Reviewer #1: N/A

Reviewer #2: Yes

3. Have the authors made all data underlying the findings in their manuscript fully available?

Reviewer #1: Yes

Reviewer #2: No

4. Is the manuscript presented in an intelligible fashion and written in standard English?

Reviewer #1: Yes

Reviewer #2: Yes

5. Review Comments to the Author

Reviewer #1: The paper titled "Risk factors for human infection with Mpox among the Mexican population" provides a comprehensive analysis of the Mpox outbreak in Mexico, identifying key risk factors associated with the disease. However, like any scientific study it may have limitations or criticisms.

The study is based on data from the Mexican Institute of Social Security (IMSS), which covers 46.3% of the Mexican population. While this is a substantial proportion, findings may not fully represent the entire population, especially those not covered by IMSS, perhaps because of different socioeconomic backgrounds. The results cannot be referred to the entire Mexican population, as the title of the paper states.

The study mentions that some data, such as sexual orientation and living with HIV status, were self-reported. Self-reported data can introduce bias due to underreporting or misreporting, especially if it concerns sexual orientation and HIV status, affecting the accuracy of the associations found.

The study excluded cases without laboratory confirmation. While this strengthens the reliability of confirmed cases, it may also omit cases with clinical symptoms of mpox but lacking laboratory confirmation, potentially underestimating the actual number of cases and the full spectrum of disease presentation.

Given the observational nature of the study, it can identify associations but cannot establish causality between risk factors and Mpox infection. Longitudinal studies would be necessary to understand the temporal sequence of exposure and infection better.

While the study performed multivariate logistic regression to adjust for several variables, there might be other confounding factors not accounted for, such as socioeconomic status, access to healthcare, and other behavioral factors that could influence the risk of Mpox infection.

The paper briefly mentions the importance of vaccination and epidemiological surveillance but does not delve deeply into specific strategies or challenges related to vaccine distribution, acceptance, and coverage, especially among high-risk groups.

Reviewer #2: It's an important study regarding the outbreak of mpox infections. However, the manuscript could be improved with some additional information:

Introduction, lines 72-74: it would be more appropraite to report the number of cases and deaths at time of declaring the global health emergency in july 2023 with the most recent update available on the epidemiology of the disease.

Also line 80-84, it's more appropriate to report the most recent data

Table 1: it would be more appropriate to report the colomn % than row % as it will show the caracteristics of the confirmed cases as compared to cases without mpox, the p-value should remain the same for conclusions on statistical differences.

please also report the min and max age and the mean+sd, and age in relevant categories.

what were the symptoms reported, treatments, vaccination status, transmission mode, hospitalisation rates, complications and/or deaths ?

Results on cumulative incidence, it would be interesting to show age and sex adjusted rates

Table III: multivariate logistic regression

- What was the total sample used

- What was the dependent variable and the size of each group in the analysis

- Were there any missing values for some variables

- Please show all the crude OR and 95% CI

- The age would be more appropriate in categories rather than continuous in this context

- Gender and sexual orientation are likely to have a colinearity in multivariable model, was this tested ?

- What was the reference category for each underlying condition, how many patients had more than one condition ?

- Diabetes seems to be associated with a lower risk of mpox infection, how is this explained

The conclusion of the manuscript and the abstract is shallow.

6. PLOS authors have the option to publish the peer review history of their article (what does this mean?). If published, this will include your full peer review and any attached files.

Reviewer #1: **Yes: **Nicola Abrescia

Reviewer #2: No

---

## [Author Response · Author response to Decision Letter 0]

27 Mar 2024

Firstly, all the authors would like to thank you for your time as reviewers, and we have responded to each of your kind inquiries. We believe that thanks to your feedback, this work has improved to achieve the set objectives.

Journal Requirements:

When completing the data availability statement of the submission form, you indicated that you will make your data available on acceptance. We strongly recommend all authors decide on a data sharing plan before acceptance, as the process can be lengthy and hold up publication timelines.

Answer: In response to your suggestion, we have placed the available data in an open repository and the link is noted in the manuscript. https://doi.org/10.6084/m9.figshare.25481266.v1

Additional Editor Comments:

The Editor's Comments:

1- Why was the HR caclulated as an epidemiological measure? In your case, the best measure would be the cumulative incidence rate, known as the incidence proportion over the specified time period.

Answer: 

Hazard Ratio (HR) is often calculated as an epidemiological measure because it provides valuable insights into the relative risk or likelihood of an event occurring over time. In many epidemiological studies, particularly those researchers are interested in understanding how certain exposures or interventions affect the risk of developing a particular outcome or disease.

The HR compares the hazard rates, or instantaneous rates of experiencing the event of interest, between different groups. It's commonly used in studies where the time to event data is important, such as in examining the impact of risk factors on disease incidence.

However, when considering the best measure for a specific case, such as the one you mentioned, the cumulative incidence rate, also known as the incidence proportion over a specified time period, might be more appropriate. This measure provides a straightforward understanding of the proportion of individuals who experience the outcome of interest within a given time frame, without considering the time to event aspect.

Therefore, we have removed the Hazard Ratio (HR) from the manuscript.

2- Given the large sample size, i would suspect that age be normally distributed. Why was the age presented as median (IQR)? I would advise the authors to provide age in different forms (continuous data, categorical data)

Answer: 

The variable age, by the Kolmogorov-Smirnov hypothesis test, had a non-normal distribution. (We attach the results).

The interquartile range (IQR) is primarily used in statistics to describe the dispersion of a dataset. The interquartile range is useful in various contexts, especially when the data do not follow a normal distribution.

We have followed your recommendation by presenting additionally the data in a categorical manner.

Age

3- There is lack of information on how the regression analysis was attempted? Which variables were included in the univariate model? and how was the selected of the variables imported in the multivariable model made?

Answer: Thank you for your feedback and inquiry regarding the regression analysis in our study. We understand the importance of transparency in reporting our methods.

In our regression analysis, we initially performed univariate modeling, which involved assessing the relationship between each potential predictor variable and the outcome variable individually. This step allowed us to identify variables that showed significant associations with the outcome. This analysis is available in Supplementary Material 1.

Regarding the selection of variables for the multivariable model, we employed a systematic approach. We considered variables that were not only statistically significant in the univariate analysis but also clinically relevant or theoretically plausible based on existing literature and biological understanding of mpox epidemiology.

4- Was model fit analysis performed? I known that the authors provided the Hosmer and Lemeshow test results; however, there are NOT consistent with the provided R-squared which reflects poor fit (which indicates that the analyzed factors do not reflect the change in the measured outcomes; there are other factors that were not accounted for). This is also supported by the wide confidence interval for variables that have a large sample size, which would normally reflect a narrow CI (like male in the multivariate model).

Answer: In the new analysis the result of the logistic regression model shows a Nagelkerke R-squared value of approximately 0.485. These values indicate that the model explains around 48.5% of the variability in the dependent variable, respectively. It's important to note that while these values provide a measure of the model's goodness of fit, they may be relatively low, suggesting that there are other variables or factors that could be influencing the dependent variable and are not included in the current model. 

Additionally, the model summary shows that the variables "Sex at birth," "Age Group," "Sexual orientation, "Condition" and some signs and symptoms are statistically significant (p < 0.05), indicating that these variables are significantly associated with the dependent variable. The confidence intervals for the regression coefficients also provide information about the accuracy of the model parameter estimates. 

In the new analysis we have gained in tightening the confidence intervals of the regression coefficients.

5- This issue is the most important of all, which is the lack of enough baseline data of included patients. Providing age, gender, sexual orientation, and HIV coinfection is not sufficient. Were not there any other data to be analyzed in your dataset? The interpretation from this dataset, given the scarcity of data, is very limited and not different from that already reported in the literature during the disease outbreak.

Answer: Our data source is the epidemiological surveillance system of mpox in Mexico. We understand that it would be beneficial to have more variables to expand our understanding of this infection. However, we provide valuable information in epidemiological terms, such as the value of cumulative incidence by key groups, which is not documented elsewhere. Additionally, quantifying the strength of association of variables is not universally established in the scientific community. By providing the value of R, we document that these variables only explain the phenomenon by 48%, thus opening a debate about which variables could provide greater insight into this disease.

6- Was multicollinearity considered? and how was it identified and dealt with?

Answer: Multicollinearity is a statistical phenomenon in which predictor variables in a logistic regression model are highly correlated. For moderate to large sample sizes, the approach to drop one of the correlated variables was established entirely satisfactory to reduce multicollinearity. On the light of different collinearity diagnostics, we may safely conclude that without increasing sample size, the second choice to omit one of the correlated variables, can reduce multicollinearity to a great extent.

Reference: Midi, H., Sarkar, S. K., & Rana, S. (2010). Collinearity diagnostics of binary logistic regression model. Journal of Interdisciplinary Mathematics, 13(3), 253–267. https://doi.org/10.1080/09720502.2010.10700699

https://www.tandfonline.com/doi/abs/10.1080/09720502.2010.10700699

After reviewing the editor's and reviewers' comments, we decided to remove the gender variable, which was clearly correlated with sex.

We also reduced categories in the sexual orientation variable, leaving only the group of men who have sex with men, which has been most strongly associated with mpox infection.

Contingency table Sex at birth * Gender

 Gender Total

 Female Male Non-binary / Bigender / Transgender / Other Unknown/ Not registered 

Sex assigned at birth Women Count 367 2 0 36 405

 % in Gender 97.9% 0.1% 0.0% 17.2% 14.3%

 Man Count 8 2228 16 173 2425

 % in Gender 2.1% 99.9% 100.0% 82.8% 85.7%

Total Count 375 2230 16 209 2830

 % in Gender 100.0% 100.0% 100.0% 100.0% 100.0%

Contingency table Sex assigned at birth * Sexual Orientation

 Sexual Orientation Total

 Heterosexual Gay/HSH Bisexual Lesbian Other Unknown/ Not registered 

Sex assigned at birth Women Count 353 1 7 3 0 41 405

 % in Sex 87.2% 0.2% 1.7% 0.7% 0.0% 10.1% 100.0%

 Man Count 744 1401 142 1 7 130 2425

 % in Sex 30.7% 57.8% 5.9% 0.0% 0.3% 5.4% 100.0%

Total Count 1097 1402 149 4 7 171 2830

 % in Sex 38.8% 49.5% 5.3% 0.1% 0.2% 6.0% 100.0%

7- In line 203, diabetes was put in the regression mode, where it was not even mentioned in the baseline variables table (Table 1). Were variables selected on purpose? You need to standardize the reporting of your data.

Answer: We have standardized the reporting of information; the diabetes variable has been removed from Table 1 and from the regression model, focusing on variables with biological plausibility and recognized by the scientific community to explain the event of interest.

8- What comorbid conditions were considered in Table 1? numerical data is required. Was HIV considered a part of these data?

The epidemiological study contains a question regarding whether the patient has any comorbidity (without specifying which), and it includes specific variables for hepatitis C, syphilis, gonorrhea, and HIV. We agree with you, and in Table 1, specific diseases are described.

Reviewers' comments:

Reviewer's Responses to Questions

Comments to the Author

Reviewer #1: The paper titled "Risk factors for human infection with Mpox among the Mexican population" provides a comprehensive analysis of the Mpox outbreak in Mexico, identifying key risk factors associated with the disease. However, like any scientific study it may have limitations or criticisms.

The study is based on data from the Mexican Institute of Social Security (IMSS), which covers 46.3% of the Mexican population. While this is a substantial proportion, findings may not fully represent the entire population, especially those not covered by IMSS, perhaps because of different socioeconomic backgrounds. The results cannot be referred to the entire Mexican population, as the title of the paper states.

Answer: Indeed, it cannot be extrapolated to the Mexican population. Therefore, following your comment, the title and interpretation of the results were adjusted to refer to the population with social security in Mexico. 

The study mentions that some data, such as sexual orientation and living with HIV status, were self-reported. Self-reported data can introduce bias due to underreporting or misreporting, especially if it concerns sexual orientation and HIV status, affecting the accuracy of the associations found.

Answer: Given that this is a study based on epidemiological information and, in general, on medicine, pathological histories are directly queried to the patient and rely on the trust in the doctor-patient relationship to obtain information. However, we acknowledge the information bias you mentioned, so much so that we decided to report it in the methodology. To ensure that the reader takes this observation into account, in the limitations section of the study, we added the following:

This study supplies an overview of mpox infection in the medical-insured population of Mexico. Due to the study's observational nature, we want to clarify that the information comes from a self-reported questionnaire answered by the individuals taking part in the survey, so it is not unbiased, and that the presence of diseases such as gonorrea, syphilis or living with HIV has not been confirmed as a part of the study.

The study excluded cases without laboratory confirmation. While this strengthens the reliability of confirmed cases, it may also omit cases with clinical symptoms of mpox but lacking laboratory confirmation, potentially underestimating the actual number of cases and the full spectrum of disease presentation.

Answer: You are correct in the observation you make; therefore, we have stated in a paragraph the fact that the incidence of cases may be underestimated.

“This study excluded suspected cases that did not have laboratory results, accounting for 4.3% of the suspected cases. Therefore, the incidence of mpox may be underestimated”. 

Given the observational nature of the study, it can identify associations but cannot establish causality between risk factors and Mpox infection. Longitudinal studies would be necessary to understand the temporal sequence of exposure and infection better.

Answer: We agree that causality cannot be established with this study but does not achieve this objective.

While the study performed multivariate logistic regression to adjust for several variables, there might be other confounding factors not accounted for, such as socioeconomic status, access to healthcare, and other behavioral factors that could influence the risk of Mpox infection.

Answer: Acknowledging the validity of the concern raised. Despite the multivariate logistic regression conducted to adjust for various variables, it is true that there may still be other confounding factors not considered, such as socioeconomic status, access to healthcare, and additional behavioral factors that could potentially influence the risk of Mpox infection. These factors indeed play significant roles in epidemiological studies and could impact the interpretation of results. Therefore, it's important to acknowledge these limitations and recognize that further research or adjustments may be necessary to better understand the true relationship between the variables studied and the risk of Mpox infection. 

In the discussion, we have acknowledged what you mention as follows: “We acknowledge that the absence of variables such as socioeconomic status, region or place of residence, and sexual behavior, including the number of sexual partners and casual sexual contacts, is a limitation of our study. These factors could potentially influence the risk of mpox infection”.

The paper briefly mentions the importance of vaccination and epidemiological surveillance but does not delve deeply into specific strategies or challenges related to vaccine distribution, acceptance, and coverage, especially among high-risk groups.

Answer: the manuscript was complemented as suggested.

Reviewer #2: It's an important study regarding the outbreak of mpox infections. However, the manuscript could be improved with some additional information:

Introduction, lines 72-74: it would be more appropraite to report the number of cases and deaths at time of declaring the global health emergency in july 2023 with the most recent update available on the epidemiology of the disease. Also line 80-84, it's more appropriate to report the most recent data

Answer: The manuscript was complemented by describing the global epidemiological situation of mpox at the time a PHEIC was declared and with the most recent data published by WHO. The epidemiological situation in Mexico was also complemented with the most recent update.

Table 1: it would be more appropriate to report the colomn % than row % as it will show the caracteristics of the confirmed cases as compared to cases without mpox, the p-value should remain the same for conclusions on statistical differences. Please also report the min and max age and the mean+sd, and age in relevant categories. what were the symptoms reported, treatments, vaccination status, transmission mode, hospitalisation rates, complications and/or deaths ?

Results on cumulative incidence, it would be interesting to show age and sex adjusted rates. 

Answer: Thank you for these comments, we made the change of % in columns in table 1 to observe the percentage of each independent variable between the confirmed cases and between the cases without mpox, what indeed do not change the p values.

Regarding other statistical measures of the age variable, because the age variable has a non-normal distribution, in Table 1 we presented the median and the interquartile range, as the most appropriate measures of central tendency and dispersion. The age variable was grouped into categories by five-year periods and is also presented in

---

## [Decision Letter · Decision Letter 1]

22 Jul 2024

PONE-D-24-01590R1Risk factors for human infection with Mpox among the Mexican population with social security.PLOS ONE

Dear Dr. Arriaga Nieto,

Thank you for submitting your manuscript to PLOS ONE. After careful consideration, we feel that it has merit but does not fully meet PLOS ONE’s publication criteria as it currently stands. Therefore, we invite you to submit a revised version of the manuscript that addresses the points raised during the review process.

We look forward to receiving your revised manuscript.

Kind regards,

Moises Leon Juarez

Academic Editor

PLOS ONE

Additional Editor Comments:

Dear Dr. Arriaga, We inform you of your work entitled “Risk factors for human infection with Mpox among the Mexican population.” The peer review process has ended. According to the reviewers' comments and observations, we consider that his work requires Major revision which is why we mention these observations below. We await his response to continue with the process.

The manuscript “Risk factors for human infection with Mpox among the Mexican population” presents a relevant topic and has the potential to contribute significantly to the field. However, the revised version submitted by the authors still contains important issues that need to be addressed for the manuscript to be suitable for the PLOS ONE audience.

Specific comments

Review the English. There are issues with the language and writing that hinder understanding and compromise the readability and flow.

Abstract: The "Materials and Methods" section should include information about the incidence and parameters of the models conducted. The results presented must be supported by the methods.

Consider including more references in the introduction. Each paragraph makes multiple assertions but only cites one reference per paragraph. Why is this the case?

The introduction needs reorganization. In the first paragraph, the authors start by discussing the 2022 outbreak. In the second and third paragraphs, they describe the transmission mode of the virus, including the hypothesis of sexual transmission relevant to the recent outbreak (2022) but not present in the description of previous cases (before 2022). In the fourth and fifth paragraphs, they discuss the clinical presentation and symptoms. In the sixth paragraph, they revisit (to some extent) the content of the first paragraph. Thus, reorganization is necessary to eliminate repeated sections and improve the coherence of the historical context and background presented.

It would be worthwhile to include the origin and historical context of mpox, its endemic regions, and how cases were previously described. When discussing the 2022 outbreak, provide temporal context, as well as epidemiological and clinical characteristics that differ from previous cases. It is necessary to contextualize the research problem and the current state of the art for the reader.

The last sentence of the final paragraph of the introduction could be moved to the methodology section. The aim at the end of the introduction differs from the aim at the beginning of the methods section. Ensure consistency (consider the abstract as well).

At the beginning of the methods section, the authors should briefly explain the health system and the IMSS, including the coverage percentage (currently listed in the last sentence of the introduction). It is not clear whether the IMSS system and the National Epidemiological Surveillance System were linked or if the notifications from NESS are part of IMSS.

In the study population section, the authors mention that they included probable and confirmed cases of mpox. Later, they state that they calculated the incidence considering the population covered by IMSS. This is confusing. The study population for each analysis conducted must be clearly defined for the reader.

The authors do not specify the data access date. Especially for studies using secondary and surveillance data, the access date must be provided as these data are continuously updated by the responsible entities.

In the definitions section, a new unnamed category is introduced in the second paragraph, where the authors state that these cases “were also studied for mpox.” What does this mean? In which category do these cases fall?

It is unnecessary to explain sexual orientation definition, just indicate whether it was self-reported (hopefully!) and what categories were included in the study. Gender identity is a separate variable from sexual orientation. The explanation in the methods section is unclear, making it difficult to understand whether these variables were analyzed separately, as they should be.

What are confirmatory studies? Are these the tests conducted to confirm mpox cases? If so, this should be defined in the definitions section. For all patients classified as suspected/probable and who provided samples, was PCR conducted for case confirmation?

How were the other variables studied obtained? It is important to describe all variables, including their origin, method of collection (self-reported?), temporal reference, and response categories (when applicable).

Explain how was calculated the incidence. What is the incidence reference? 10,000 people? This needs to be described in the methods.

The authors refer to “key populations” without defining what it is and how this variable was created to estimate the stratified incidence.

In the univariate model, what did the authors consider as “strength of association”? (line 184). The described method should be in the main text.

The paragraph (lines 183 to 189) and the following paragraph (lines 190-197) contain repetitions. Review these sections. What parameters were used in the multivariate model? Variables described here appear for the first time in the text without explanations.

Ensure consistency of variables presented in the dataset, main text, and tables. For example, the variable “gender” appears under different denominations and different categorizations in the main text, tables and dataset.

Figure S1 is dispensable. Table S1 needs to be formatted according to the journal's formatting and style guidelines. Adjust the titles.

Tables S1 and S2 should be in the same document, in Word/PDF format, following the journal's table formatting rules.

Figure 1 (sample composition flowchart) is cited incorrectly at the end of the methods section.

In the results, the authors present stratified results by confirmed (by PCR) and unconfirmed cases, which was not described in the methods.

In Figure 2, it would be more informative to present the positivity rate per month (% of positives among those tested). In its current form, the figure title is inadequate.

Only in the description of the results in Table 3 do we learn that the incidence rate presented is per 100,000 IMSS beneficiaries. This needs to be described in the methods and table title.

The multivariate model results are presented in a segmented manner. Review this to create a cohesive text with logical flow.

The first paragraph of the discussion presents an objective conflicting with those in the abstract, the end of the introduction, and the beginning of the methods section. The first paragraph of the discussion should provide an overview of the main study findings, which will be discussed further.

“The discovery of mpox transmission” does not seem an appropriate term.

The second paragraph of the discussion should be in the introduction/repeats content already in the introduction without adding value to the discussion. I suggest removing it.

How the authors created the category “bigender", presented at the discussion?

There seems to be some difficulty on the part of the authors regarding the variables of sex, gender identity, and sexual orientation. I suggest reviewing studies that have clearly addressed these variables. https://www.thelancet.com/journals/lanam/article/PIIS2667-193X(22)00223-X/fulltext

https://www.thelancet.com/journals/lancet/article/PIIS0140-6736(23)00273-8/fulltext

The authors present results for confirmed cases but refer back to probable cases in the conclusions. The conclusion needs to be revised to address the study's proposed aims.

All figures and tables need to be reviewed for adherence to formatting standards and appropriate titles.

Reviwer 2

1) Improve the English wording

2) Lack of congruence between the objective of the study, methodology and results. It seems that it is looking for an incidence, but it is not a cohort. We suggest reviewing the methodology when you are looking for description of the characteristics and associated factors, we suggest a cross-sectional study with prevalence and associated factors.

3) Improve language, avoid the word gay, only MSM, use inclusive language.

4) Table 1.- two columns are not necessary.

5) The results are reported in means with IQR, but due to the size of the sample it is worth looking at its distribution.

6) The OR data are poorly written, it should be as follows: MSM OR 8.28 (95%CI 6.50 – 10.55), p<0.001; the semicolon is used to separate ideas

7) 207-208 eliminate, emphasize the main results of your research

8) 210-211 Seems like information is repeated. 212 The terminology “American House” is not well understood. 215 Do we have the complete data to affirm Mexico is the fourth country in America with Mpox cases?

9) Discussion could improve if the author compares local data with other similar studies in other countries. 223: We suggest that cumulative incidence could not be reported with de current methodology

10) Idea of 225-226 is not understood.

13) The conclusion does not answer the main objective.

14) It is difficult to consider a descriptive study

Reviewers' comments:

Reviewer's Responses to Questions

**Comments to the Author**

1. If the authors have adequately addressed your comments raised in a previous round of review and you feel that this manuscript is now acceptable for publication, you may indicate that here to bypass the “Comments to the Author” section, enter your conflict of interest statement in the “Confidential to Editor” section, and submit your "Accept" recommendation.

Reviewer #3: (No Response)

Reviewer #4: (No Response)

2. Is the manuscript technically sound, and do the data support the conclusions?

Reviewer #3: Partly

Reviewer #4: Partly

3. Has the statistical analysis been performed appropriately and rigorously? 

Reviewer #3: I Don't Know

Reviewer #4: No

4. Have the authors made all data underlying the findings in their manuscript fully available?

Reviewer #3: Yes

Reviewer #4: Yes

5. Is the manuscript presented in an intelligible fashion and written in standard English?

Reviewer #3: No

Reviewer #4: No

6. Review Comments to the Author

Reviewer #3: General comments

The manuscript “Risk factors for human infection with Mpox among the Mexican population” presents a relevant topic and has the potential to contribute significantly to the field. However, the revised version submitted by the authors still contains important issues that need to be addressed for the manuscript to be suitable for the PLOS ONE audience.

Specific comments

Review the English. There are issues with the language and writing that hinder understanding and compromise the readability and flow.

Abstract: The "Materials and Methods" section should include information about the incidence and parameters of the models conducted. The results presented must be supported by the methods.

Consider including more references in the introduction. Each paragraph makes multiple assertions but only cites one reference per paragraph. Why is this the case?

The introduction needs reorganization. In the first paragraph, the authors start by discussing the 2022 outbreak. In the second and third paragraphs, they describe the transmission mode of the virus, including the hypothesis of sexual transmission relevant to the recent outbreak (2022) but not present in the description of previous cases (before 2022). In the fourth and fifth paragraphs, they discuss the clinical presentation and symptoms. In the sixth paragraph, they revisit (to some extent) the content of the first paragraph. Thus, reorganization is necessary to eliminate repeated sections and improve the coherence of the historical context and background presented.

It would be worthwhile to include the origin and historical context of mpox, its endemic regions, and how cases were previously described. When discussing the 2022 outbreak, provide temporal context, as well as epidemiological and clinical characteristics that differ from previous cases. It is necessary to contextualize the research problem and the current state of the art for the reader.

The last sentence of the final paragraph of the introduction could be moved to the methodology section. The aim at the end of the introduction differs from the aim at the beginning of the methods section. Ensure consistency (consider the abstract as well).

At the beginning of the methods section, the authors should briefly explain the health system and the IMSS, including the coverage percentage (currently listed in the last sentence of the introduction). It is not clear whether the IMSS system and the National Epidemiological Surveillance System were linked or if the notifications from NESS are part of IMSS.

In the study population section, the authors mention that they included probable and confirmed cases of mpox. Later, they state that they calculated the incidence considering the population covered by IMSS. This is confusing. The study population for each analysis conducted must be clearly defined for the reader.

The authors do not specify the data access date. Especially for studies using secondary and surveillance data, the access date must be provided as these data are continuously updated by the responsible entities.

In the definitions section, a new unnamed category is introduced in the second paragraph, where the authors state that these cases “were also studied for mpox.” What does this mean? In which category do these cases fall?

It is unnecessary to explain sexual orientation definition, just indicate whether it was self-reported (hopefully!) and what categories were included in the study. Gender identity is a separate variable from sexual orientation. The explanation in the methods section is unclear, making it difficult to understand whether these variables were analyzed separately, as they should be.

What are confirmatory studies? Are these the tests conducted to confirm mpox cases? If so, this should be defined in the definitions section. For all patients classified as suspected/probable and who provided samples, was PCR conducted for case confirmation?

How were the other variables studied obtained? It is important to describe all variables, including their origin, method of collection (self-reported?), temporal reference, and response categories (when applicable).

Explain how was calculated the incidence. What is the incidence reference? 10,000 people? This needs to be described in the methods.

The authors refer to “key populations” without defining what it is and how this variable was created to estimate the stratified incidence.

In the univariate model, what did the authors consider as “strength of association”? (line 184). The described method should be in the main text.

The paragraph (lines 183 to 189) and the following paragraph (lines 190-197) contain repetitions. Review these sections. What parameters were used in the multivariate model? Variables described here appear for the first time in the text without explanations.

Ensure consistency of variables presented in the dataset, main text, and tables. For example, the variable “gender” appears under different denominations and different categorizations in the main text, tables and dataset.

Figure S1 is dispensable. Table S1 needs to be formatted according to the journal's formatting and style guidelines. Adjust the titles.

Tables S1 and S2 should be in the same document, in Word/PDF format, following the journal's table formatting rules.

Figure 1 (sample composition flowchart) is cited incorrectly at the end of the methods section.

In the results, the authors present stratified results by confirmed (by PCR) and unconfirmed cases, which was not described in the methods.

In Figure 2, it would be more informative to present the positivity rate per month (% of positives among those tested). In its current form, the figure title is inadequate.

Only in the description of the results in Table 3 do we learn that the incidence rate presented is per 100,000 IMSS beneficiaries. This needs to be described in the methods and table title.

The multivariate model results are presented in a segmented manner. Review this to create a cohesive text with logical flow.

The first paragraph of the discussion presents an objective conflicting with those in the abstract, the end of the introduction, and the beginning of the methods section. The first paragraph of the discussion should provide an overview of the main study findings, which will be discussed further.

“The discovery of mpox transmission” does not seem an appropriate term.

The second paragraph of the discussion should be in the introduction/repeats content already in the introduction without adding value to the discussion. I suggest removing it.

How the authors created the category “bigender", presented at the discussion?

There seems to be some difficulty on the part of the authors regarding the variables of sex, gender identity, and sexual orientation. I suggest reviewing studies that have clearly addressed these variables. https://www.thelancet.com/journals/lanam/article/PIIS2667-193X(22)00223-X/fulltext

https://www.thelancet.com/journals/lancet/article/PIIS0140-6736(23)00273-8/fulltext

The authors present results for confirmed cases but refer back to probable cases in the conclusions. The conclusion needs to be revised to address the study's proposed aims.

All figures and tables need to be reviewed for adherence to formatting standards and appropriate titles.

Reviewer #4: 1) Improve the English wording

2) Lack of congruence between the objective of the study, methodology and results. It seems that it is looking for an incidence, but it is not a cohort. We suggest reviewing the methodology when you are looking for description of the characteristics and associated factors, we suggest a cross-sectional study with prevalence and associated factors.

3) Improve language, avoid the word gay, only MSM, use inclusive language.

4) Table 1.- two columns are not necessary.

5) The results are reported in means with IQR, but due to the size of the sample it is worth looking at its distribution.

6) The OR data are poorly written, it should be as follows: MSM OR 8.28 (95%CI 6.50 – 10.55), p<0.001; the semicolon is used to separate ideas

7) 207-208 eliminate, emphasize the main results of your research

8) 210-211 Seems like information is repeated. 212 The terminology “American House” is not well understood. 215 Do we have the complete data to affirm Mexico is the fourth country in America with Mpox cases?

9) Discussion could improve if the author compares local data with other similar studies in other countries. 223: We suggest that cumulative incidence could not be reported with de current methodology

10) Idea of 225-226 is not understood.

13) The conclusion does not answer the main objective.

14) It is difficult to consider a descriptive study

7. PLOS authors have the option to publish the peer review history of their article (what does this mean?). If published, this will include your full peer review and any attached files.

Reviewer #3: No

Reviewer #4: No

---

## [Author Response · Author response to Decision Letter 1]

22 Oct 2024

Dear Reviewers, we greatly appreciate your comments. We have addressed each of them.

Reviewer 1.

1. Review the English. There are issues with the language and writing that hinder understanding and compromise the readability and flow.

A=We have carefully reviewed the language and writing in the manuscript.

2. Abstract: The "Materials and Methods" section should include information about the incidence and parameters of the models conducted. The results presented must be supported by the methods.

A= Thank you for your comment, we carefully reviewed the abstract section to ensure it was consistent with the manuscript, including the materials and methods section where we added the cumulative incidence and the models we performed.

3. Consider including more references in the introduction. Each paragraph makes multiple assertions but only cites one reference per paragraph. Why is this the case?

A= The introduction was modified, and several references were included to support the assertions.

4. The introduction needs reorganization. In the first paragraph, the authors start by discussing the 2022 outbreak. In the second and third paragraphs, they describe the transmission mode of the virus, including the hypothesis of sexual transmission relevant to the recent outbreak (2022) but not present in the description of previous cases (before 2022). In the fourth and fifth paragraphs, they discuss the clinical presentation and symptoms. In the sixth paragraph, they revisit (to some extent) the content of the first paragraph. Thus, reorganization is necessary to eliminate repeated sections and improve the coherence of the historical context and background presented.

A= The introduction was reorganized, repeated sections were removed, and background and historical events were arranged chronologically to improve the coherence of the section.

5. It would be worthwhile to include the origin and historical context of mpox, its endemic regions, and how cases were previously described. When discussing the 2022 outbreak, provide temporal context, as well as epidemiological and clinical characteristics that differ from previous cases. It is necessary to contextualize the research problem and the current state of the art for the reader.

A= The origin and historical context of mpox, as well as the description of the epidemiological and clinical features of the multinational outbreak in 2022, were added to the introduction.

6. The last sentence of the final paragraph of the introduction could be moved to the methodology section. The aim at the end of the introduction differs from the aim at the beginning of the methods section. Ensure consistency (consider the abstract as well).

A= As suggested, the last sentence of the last paragraph of the introduction was moved to the methodology section. The objective was checked to be consistent across all sections such as the end of the introduction, methodology and abstract.

7. At the beginning of the methods section, the authors should briefly explain the health system and the IMSS, including the coverage percentage (currently listed in the last sentence of the introduction). It is not clear whether the IMSS system and the National Epidemiological Surveillance System were linked or if the notifications from NESS are part of IMSS.

A= In the methods section, information was expanded to explain IMSS coverage and the link between the IMSS and the National Epidemiological Surveillance System of Mpox.

8. In the study population section, the authors mention that they included probable and confirmed cases of mpox. Later, they state that they calculated the incidence considering the population covered by IMSS. This is confusing. The study population for each analysis conducted must be clearly defined for the reader.

A= In the study population section, it is clarified that the study population was all probable cases of mpox reported by IMSS medical units in 2022-2023. In later sections, it is clarified how the cumulative incidence was calculated.

9. The authors do not specify the data access date. Especially for studies using secondary and surveillance data, the access date must be provided as these data are continuously updated by the responsible entities.

A= In the methodology, at the end of the study population section, it was specified that June 26, 2023 was the date the database was accessed.

10. In the definitions section, a new unnamed category is introduced in the second paragraph, where the authors state that these cases “were also studied for mpox.” What does this mean? In which category do these cases fall?

A= Thank you very much for your comment. In the variable definition section it was specified that these cases referred to in the commented paragraph are also probable cases of mpox.

11. It is unnecessary to explain sexual orientation definition, just indicate whether it was self-reported (hopefully!) and what categories were included in the study. Gender identity is a separate variable from sexual orientation. The explanation in the methods section is unclear, making it difficult to understand whether these variables were analyzed separately, as they should be.

A= In accordance with your comments, thank you very much. The definition of sexual orientation was removed. In the methodology section, it was specified that the sexual orientation variable was collected by self-report and the categories of the variable are described. Likewise, the gender variable was collected by self-report and the categories of the variable were described. These variables were analyzed separately.

12. What are confirmatory studies? Are these the tests conducted to confirm mpox cases? If so, this should be defined in the definitions section. For all patients classified as suspected/probable and who provided samples, was PCR conducted for case confirmation?

A= The subheading for confirmatory studies was removed. The definitions of confirmed case and ruled out case were added to the variable definitions section. It was also clarified that all patients classified as probable cases had samples taken for confirmation by PCR testing performed by authorized laboratories.

13. How were the other variables studied obtained? It is important to describe all variables, including their origin, method of collection (self-reported?), temporal reference, and response categories (when applicable).

A= In the methodology section, it was described how the other variables such as sociodemographic, epidemiological and clinical variables were collected. The definition of the variables and their categories was also added.

14. Explain how was calculated the incidence. What is the incidence reference? 10,000 people? This needs to be described in the methods.

A= In the methods section, further information was provided on how the cumulative incidence was calculated.

15. The authors refer to “key populations” without defining what it is and how this variable was created to estimate the stratified incidence.

A= The term “key populations” was eliminated since, in effect, it was not defined; rather, we refer to population subgroups and this was indicated in the statistical analysis section of the methodology. It was also described how the stratified cumulative incidence was estimated in population subgroups such as MSM and people living with HIV.

16. In the univariate model, what did the authors consider as “strength of association”? (line 184). The described method should be in the main text.

A= In the univariate model, the term “strength of association” was corrected, clarifying that it is statistical significance.

17. The paragraph (lines 183 to 189) and the following paragraph (lines 190-197) contain repetitions. Review these sections. What parameters were used in the multivariate model? Variables described here appear for the first time in the text without explanations.

A= Thank you for your comment. We have reviewed the highlighted paragraphs and removed the repeated sections, writing a clearer description of the analysis performed. It was described that the multivariate analysis was performed with the independent variables that showed statistical significance in the univariate analysis, with a level of statistical significance of p<0.05. Additionally, the variables that were indicated in this section were eliminated and included in the variable definitions and data collection section.

18. Ensure consistency of variables presented in the dataset, main text, and tables. For example, the variable “gender” appears under different denominations and different categorizations in the main text, tables and dataset.

A= Thank you for your valuable comment. We have reviewed the database, the main text, and the tables, and made corrections to the gender and sexual orientation categories. We ensured consistency across the tables, database, and manuscript text.

19. Figure S1 is dispensable. Table S1 needs to be formatted according to the journal's formatting and style guidelines. Adjust the titles.

A= Thank you very much for your comment. Figure S1 has been removed and Table S1 was formatted according to the PLOS ONE formatting and style guidelines and the title was adjusted.

20. Tables S1 and S2 should be in the same document, in Word/PDF format, following the journal's table formatting rules.

A= Thank you very much for your comment. Table S1 were inserted into a Word sheet that was converted to a PDF document and Table S2 was removed.

21. Figure 1 (sample composition flowchart) is cited incorrectly at the end of the methods section.

A= We adjusted the citation of Figure 1 according to manuscript body formatting guidelines.

22. In the results, the authors present stratified results by confirmed (by PCR) and unconfirmed cases, which was not described in the methods.

A= Thank you for your comment. The methodology section was complemented with the description of the comparative analysis carried out to evaluate the clinical symptoms and signs between patients with laboratory-confirmed mpox and those with laboratory-ruled mpox; the statistical test and level of significance used are indicated.

23. In Figure 2, it would be more informative to present the positivity rate per month (% of positives among those tested). In its current form, the figure title is inadequate.

A= Thank you for your comment, we gladly accept this suggestion and present in Figure 2 the mpox positivity rate per month and change the figure title to a more appropriate one.

24. Only in the description of the results in Table 3 do we learn that the incidence rate presented is per 100,000 IMSS beneficiaries. This needs to be described in the methods and table title.

A= Thank you very much for your comment. We have included in the methodology section the complete description of how the incidence rate per 100,000 IMSS beneficiaries was calculated; we also included this description in the titles and tables where we present the incidence rate.

25. The multivariate model results are presented in a segmented manner. Review this to create a cohesive text with logical flow.

A= Thank you for your comment, the segmented way of presenting the results of the multivariate analysis has been changed, creating a more fluid text.

26. The first paragraph of the discussion presents an objective conflicting with those in the abstract, the end of the introduction, and the beginning of the methods section. The first paragraph of the discussion should provide an overview of the main study findings, which will be discussed further.

A= Thank you for your comment, there was indeed a mismatch between the first paragraph of the discussion and the sections indicated, so the first paragraph of the discussion was changed to provide an overview of the main findings of the study.

27. “The discovery of mpox transmission” does not seem an appropriate term.

A= Thank you very much for your comment, the entire paragraph that included this inappropriate term was removed from the discussion in accordance with the following comment.

28. The second paragraph of the discussion should be in the introduction /repeats content already in the introduction without adding value to the discussion. I suggest removing it.

A= Thank you for your comment, we agree with your suggestion and removed the paragraph from the discussion as it contained repetitions of the description of the evolution of the multinational mpox outbreak, and effectively did not add value to the discussion.

29. How the authors created the category “bigender", presented at the discussion?

A= The gender variable was collected according to the categories established in the standardized mpox CRF provided by the national mpox epidemiological surveillance system. The bigender category was one of the categories collected directly from the patient through the questionnaire. Since no patient (0 cases) was reported in this bigender category and based on the review of the articles you recommended to us on the classification of the gender variable, we eliminated this category without affecting the analysis.

30. There seems to be some difficulty on the part of the authors regarding the variables of sex, gender identity, and sexual orientation. I suggest reviewing studies that have clearly addressed these variables. https://www.thelancet.com/journals/lanam/article/PIIS2667-193X(22)00223-X/fulltext 

https://www.thelancet.com/journals/lancet/article/PIIS0140-6736(23)00273-8/fulltext

A= Thank you very much for your suggestions, we reviewed the recommended articles, and the gender and sexual orientation variables were corrected in the different sections of the manuscript.

31. The authors present results for confirmed cases but refer back to probable cases in the conclusions. The conclusion needs to be revised to address the study's proposed aims.

A= The discussion and conclusions section has been modified so that it no longer refers to probable cases.

32. All figures and tables need to be reviewed for adherence to formatting standards and appropriate titles.

A= Thank you for your comment, all tables were reviewed, and the format was adjusted according to the PLOS ONE formatting and style guidelines. Table titles have also been adjusted.

Reviwer 2

1) Improve the English wording

A= We have carefully reviewed the language and writing in the manuscript.

33. Lack of congruence between the objective of the study, methodology and results. It seems that it is looking for an incidence, but it is not a cohort. We suggest reviewing the methodology when you are looking for description of the characteristics and associated factors, we suggest a cross-sectional study with prevalence and associated factors.

A= Thank you very much for your comment. Since all the cases were new cases identified in the IMSS medical units and reported to the national epidemiological surveillance system, we calculated the cumulative incidence per 100,000 IMSS beneficiaries.

Regarding the type of study, indeed, in the methodology we describe that we conducted a retrospective observational study using a cross-sectional survey to assess mpox cumulative incidence and associated factors.

34. Improve language, avoid the word gay, only MSM, use inclusive language.

A=The language was improved by using inclusive language by eliminating the word gay and using only the term MSM throughout the manuscript.

35. Table 1.- two columns are not necessary.

A= Based on your comment, the second column of Table 1 was removed.

36. The results are reported in means with IQR, but due to the size of the sample it is worth looking at its distribution.

A= The variable age, by the Kolmogorov-Smirnov hypothesis test, had a non-normal distribution. (We attach the results in response to reviewers document).

The interquartile range (IQR) is primarily used in statistics to describe the dispersion of a dataset. The interquartile range is useful in various contexts, especially when the data do not follow a normal distribution.

37. The OR data are poorly written, it should be as follows: MSM OR 8.28 (95%CI 6.50 – 10.55), p<0.001; the semicolon is used to separate ideas

A=We have revised the data presentation as suggested, including the appropriate use of the semicolon to

---

## [Editor Report · Decision Letter 2]

30 Oct 2024

Risk factors for human infection with Mpox among the Mexican population with social security.

PONE-D-24-01590R2

Dear Dr. Arriaga Nieto,

We’re pleased to inform you that your manuscript has been judged scientifically suitable for publication and will be formally accepted for publication once it meets all outstanding technical requirements.

Kind regards,

Moises Leon Juarez

Academic Editor

PLOS ONE

Additional Editor Comments (optional):

The authors responded and made the changes recommended by the reviewers, so that the article is ready to be published in the journal.
---

## [Editor Report · Acceptance letter]

6 Nov 2024

PONE-D-24-01590R2 

PLOS ONE

Dear Dr. Arriaga Nieto, 

I'm pleased to inform you that your manuscript has been deemed suitable for publication in PLOS ONE. Congratulations! Your manuscript is now being handed over to our production team.

Kind regards, 

on behalf of

Dr. Moises Leon Juarez 

Academic Editor

PLOS ONE